# Detection by Real-Time PCR of *Helicobacter pylori* and Clarithromycin Resistance Compared to Histology on Gastric Biopsies

**DOI:** 10.3390/microorganisms12112192

**Published:** 2024-10-30

**Authors:** Guillaume Pittie, Terry Laurent, Jean Radermacher, Sophie Herens, Anca Boeras, Giang Ho

**Affiliations:** 1Clinical Microbiology Department, CHC MontLégia, 4000 Liège, Belgium; terry.laurent@chc.be (T.L.); sophie.herens@chc.be (S.H.); anca.boeras@chc.be (A.B.); giang.ho@chc.be (G.H.); 2Pathological Anatomy and Cytology Laboratory, CHC MontLégia, 4000 Liège, Belgium

**Keywords:** *Helicobacter pylori*, real-time PCR, histology

## Abstract

The global rise in *Helicobacter pylori* (*H. pylori*)-related gastric complications is largely driven by increasing antimicrobial resistance and treatment failures. As a result, accurate diagnosis followed by effective treatment is crucial. We analyzed 232 gastric biopsy samples from patients undergoing endoscopy during the method validation phase, followed by 502 samples in the routine evaluation phase. Each sample was tested using the Allplex™ *H. pylori* and ClariR Assay on a CFX96™ real-time PCR (RT-PCR) system, with results processed through Seegene Viewer software. In the validation phase, RT-PCR results were compared to bacterial culture, while in the routine phase, they were compared to histology. The sensitivity and specificity for *H. pylori* detection were 100% and 96.05% (95% Confidence Interval [CI]: 93.38–98.73), respectively. For clarithromycin resistance detection, the sensitivity and specificity were 100% and 93.33% (95% CI: 84.4–100). Additionally, RT-PCR identified 11 positive samples (10.89%) that histology failed to detect. Incorporating the Allplex™ *H. pylori* and ClariR Assay into our laboratory workflow improved efficiency, reduced turnaround time (TaT), and proved to be more sensitive than both culture and histology combined.

## 1. Introduction

*Helicobacter pylori* (*H. pylori*) is a microaerophilic, spiral-shaped, gram-negative bacterium [1,2]. Due to its ability to colonize and chronically infect the human stomach [1,2], this pathogen has been identified as a major etiological factor in the development of gastric adenocarcinoma and is thought to be responsible for more cases of cancer worldwide than hepatitis B and C viruses combined [3].

Indeed, *H. pylori* was classified as a Group 1 carcinogen in 1994 [4]. Since this classification, several studies have demonstrated this association between chronic *H. pylori* infection and gastric cancer, supporting the importance of *H. pylori* eradication in the prevention of gastric cancer [5,6,7,8].

According to recent studies, the prevalence of *H. pylori* infection in Europe varies between 20 and 40%, with a decrease in the number of cases in recent years [9,10,11]. However, gastric cancer remains an important cancer worldwide, responsible for more than one million new cases in 2020 and an estimated 769,000 deaths (one in 13 deaths worldwide), ranking fifth in incidence and fourth in mortality worldwide [12]. With 89% of non-cardia gastric cancers caused by *H. pylori*, representing 78% of all gastric cancer cases [13], this bacterium is a global burden for the health of the world’s population.

Despite the decline in the prevalence of *H. pylori* infection, the increase in antibiotic resistance is an obstacle to the eradication of this bacterium. Especially for clarithromycin, which is a key antibiotic for the treatment of *H. pylori* [14]. A study from the European *Helicobacter pylori* Antimicrobial Susceptibility Testing Working Group Resistance showed that clarithromycin resistance was increasing every year, coming from 17.5% in 2008 to 21.4% in 2018 [15]. In this study, clarithromycin had a higher resistance rate compared to levofloxacin (15.8%), rifampicin (0.9%), amoxicillin (0.2%), and tetracycline (0%). Only metronidazole showed a higher resistance rate (38.9%) [15], but its impact on *H. pylori* eradication is limited [16].

Numerous studies have established a connection between point mutations in the peptidyl transferase loop of the V domain in the 23S rRNA gene and clarithromycin resistance in clinical *H. pylori* strains from various geographic regions [17,18,19]. The peptidyl-transferase region is essential for antibiotic binding, and mutations in this region disrupt the binding of macrolides and ribosomal subunits, leading to bacterial resistance to these antibiotics [20].

The most frequently observed and well-characterized mutations in *H. pylori* are located at two adjacent nucleotide positions: an adenine-to-guanine transition at position 2142 (*A2142G*) or 2143 (*A2143G*). Less commonly, an adenine-to-cytosine transversion at position 2142 (*A2142C*) occurs. These mutations are responsible for more than 90% of clarithromycin resistance in developed countries [21]. In particular, the mutation at position 2143 is often linked to varying degrees of resistance (with MICs ranging from 2 to 256 mg/L), while the mutation at position 2142 generally results in a more limited resistance (MIC of 64 mg/L) [22].

Additionally, several other point mutations have been identified, including *A2115G*, *G2141A*, *T2117C*, *T2182C*, *T2289C*, *G224A*, *C2245T*, and *C2611A*. Although these mutations are less common, the clinical relevance of mutations such as *A2115G*, *G2141A*, *T2117C*, *T2289C*, *G224A*, and *C2245T* is still uncertain, as their roles have not been consistently documented [23,24,25]. However, the *T2182C* and *C2611A* mutations have been linked to lower resistance levels [22,25].

To avoid useless treatment, induction of resistance to other bacteria, or adverse events due to quadruple therapy [26,27], the Maastricht VI consensus advises that clarithromycin susceptibility should at least be tested by standard methods, including *H. pylori* culture and susceptibility testing, or, if available, by molecular techniques before prescribing any clarithromycin-containing therapy. Indeed, molecular tests such as real-time PCR (RT-PCR) are now commercially available and provide excellent sensitivity and specificity for detecting both *H. pylori* and its clarithromycin susceptibility [28,29].

Currently, *H. pylori* infection can be diagnosed through various methods, including non-invasive tests that do not require endoscopy or biopsy samples, such as antibody detection in serum, saliva, or urine, the urea breath test (UBT), stool antigen test (SAT), and PCR from stool samples. Invasive tests, which require biopsy samples obtained via endoscopy, include histopathology, the rapid urease test (RUT), culture, and PCR from biopsy specimens [30,31]. The choice of diagnostic method depends on factors such as the patient’s age, the presence of alarm signs or symptoms, use of non-steroidal anti-inflammatory drugs, as well as local availability, test accuracy, and cost [32].

Before May 2023, we diagnosed *H. pylori* infection through culture and histological examination. Antibiotic susceptibility testing was only conducted after two treatment failures.

Given the epidemiological shifts and advancements in diagnostics in recent years, we decided to revise our approach in the laboratory. We implemented an RT-PCR assay to detect *H. pylori* and identify the three most common point mutations associated with clarithromycin resistance (*A2142C*, *A2142G*, and *A2143G*). To initially assess the effectiveness of this method, we compared it with the culture of gastric biopsies.

After eight months of routine use, we evaluated the added value of this PCR assay in comparison to the histological analysis of gastric biopsies.

## 2. Materials and Methods

### 2.1. Biopsies Collection

For the comparison between RT-PCR and bacterial culture, 232 gastric biopsies were obtained from patients submitted to endoscopy for various gastric conditions (dyspepsia, peptic ulcer, precancerous lesions, mucosa-associated lymphoid tissue (MALT) lymphoma) between 11 August 2022 and 16 May 2023.

For the routine evaluation of the RT-PCR and its comparison to histology, 502 gastric biopsies were obtained from patients submitted to endoscopy for various gastric conditions (dyspepsia, peptic ulcer, precancerous lesions, MALT lymphoma, …) between 23 May 2023 and 17 January 2024. 327 adults and 175 children were included in this part of the study.

The sample preparation procedure is as follows: Using a dispenser, add 500 µL of physiological saline to a 1.5 mL microtube. Using a sterile loop, pipette, or swab, transfer the biopsy fragment(s) into the microtube. Grind the biopsy with a pestle for 10–15 s, ensuring the suspension is homogeneous. Then, add 300 µL of physiological saline to complete the suspension. Homogenize the mixture by performing five cycles of suction and release. Transfer 400 µL of the solution to an Axygen^®^ (Illkirch, France) microtube, avoiding any unground tissue. This tube is then stored at −20 °C until molecular analysis. The initial microtube, containing the remaining 400 µL, is stored at −80 °C until inoculation or for further molecular testing.

### 2.2. Real-Time PCR

RT-PCR has been performed on each sample for the first part (n = 232) and the second part (n = 502) of the study.

For RT-PCR analysis, we proceed as follows: Mix 600 μL of lysis buffer with the sample in an Eppendorf. Add 30 μL of proteinase K. Vortex the Eppendorf. Incubate the Eppendorf at 55 °C ± 3 °C for 1 h. After 1 h, recover the Eppendorf and vortex again for 10 s. Take a new sterile Eppendorf and transfer 300 μL of supernatant from the incubated Eppendorf into the new Eppendorf. Lysis of the sample by adding 150 μL LB lysis buffer and 10 μL proteinase K (inactivation of nucleases and release of nucleic acids). Addition of 10 μL of magnetic beads to the sample with 586 μL of Binding Buffer BB. In this step, the beads surrounded by the attached nucleic acids are magnetized in the bottom of the wells using a magnet called the “nucleoMag SEP”. Now that the beads are held at the bottom of the wells with the nucleic acids, the washing procedure can begin without any risk of losing DNA during purification. To do the washing step, remove the primary supernatant. Add 490 μL of “Wash Buffer 1” with suction and reflux, then discard. Add 490 μL of Wash Buffer 2 containing ethanol with suction and reflux, then discard. Addition of 500 μL of “Wash Buffer 3” with suction-discharge, then elimination. Finally, the highly purified nucleic acid is eluted with 60 μL of Elution Buffer. Once the DNA of interest has been isolated and purified, the sample is placed on the Starlet instrument (Seegeene^®^ (Seoul, Republic of Korea)), which adds the reagents specific to PCR. Once the mix has been made, the plate is placed in the CFX 96 (bioRad^®^ (Hercules, CA, USA)) for the amplification and detection step.

Extraction of DNA was performed with the StarMag Universal DNA/RNA extraction kit (Seegene^®^) on the Starlet instrument (Seegene^®^). The amplification and detection of *H. pylori* and clarithromycin mutations were performed on a CFX 96 (bioRad^®^) using the Allplex *H. pylori* and ClariR Assay (Seegene^®^). The analysis of amplification data are realized with the software Seegeene Viewer V3 for real-time instruments. Amplification is performed up to 50 cycles thresholds (Ct). The kits have been used according to the manufacturer’s instructions.

### 2.3. Bacterial Culture

Bacterial cultures were conducted on each sample from the first part of the study (n = 232). In the second phase, cultures were only performed on samples that tested positive for a clarithromycin resistance gene using our RT-PCR assay (n = 26). Gastric biopsies were cultured on bioMérieux^®^ (Marcy l’Etoile, France)) pylori agar (Figure 1).

From the aliquot of shredded material frozen at −80 °C, 150 µL of suspension was placed in the center of the Pylori-selective culture medium (bioMérieux^®^) and a star-shaped isolation procedure was performed. Each culture medium is incubated under a microaerobic atmosphere in a chamber at 36 ± 1 °C. After inoculation, the presence of colonies was screened 72 h after inoculation of the biopsy on Pylori medium, and then every 48 h until day 10. The suspected colonies were identified by MALDI-TOF MS Biotyper Sirius (Bruker ^®^ (Billerica, MA, USA)), Gram staining microscopy, and biochemical reactions (oxidase, catalase, urease). The MALDI-TOF MS used library is the MBT Compass HT IVD Library (IVD-12299 Library module (2023).

To evaluate the Limit of Detection (LoD), we diluted de CCUG *Helicobacter pylori* 38771 strain from a bacterial charge of 10^6^ CFU/mL to 10 CFU/mL.

### 2.4. Antibiotic Susceptibility Testing (AST)

For the method validation, AST was performed on each culture-positive sample (n = 29). For the second part, AST was performed on every sample positive for a clarithromycin resistance gene with our RT-PCR which we were able to cultivate on the bioMérieux^®^ pylori agar.

Three McF bacterial suspensions of the *Helicobacter pylori* strain were inoculated on 4 Mueller-Hinton Fastidious (MH-F-bioMérieux^®^) agar plates. Gradient tests (Etest—bioMérieux^®^) for Levofloxacin, Metronidazole, Tetracycline, Amoxicilin, and Claritromycin were placed in the center of the plates. Minimum inhibitory concentrations (MIC) are read after 48–72 h of incubation at 36 ± 1 °C in a microaerobic atmosphere.

### 2.5. Histological Analysis

In the second part of the study (n = 502), histological analysis is performed on each sample by a pathologist. All gastric biopsies were fixed in formalin and then embedded in paraffin. Prepared glass slides were stained with hematoxylin-eosin.

We used immunohistochemical (IHC) staining in specific cases such as chronic gastritis, atrophic gastritis, extensive intestinal metaplasia, or in follow-up biopsies after eradication treatment (n = 106). IHC staining was performed using Roche^®^ (Bâle, Switzerland) Benchmark ULTRA *Helicobacter pylori* matrices (clone sp48). See Figure 2 for staining histologic analysis illustration.

### 2.6. Statistical Analysis

All de statistics have been calculated on Microsoft^®^ Excel version 16.88.

## 3. Results

### 3.1. Method Validation

In the first phase of our study, 37 out of 232 biopsies (15.94%) tested positive for *H. pylori* by PCR, with 8 of these PCR-positive samples showing negative results in culture. The PCR-positive but culture-negative samples had an average Ct value of 39.85, whereas the culture-positive samples had an average Ct value of 31.73. Among the positive samples, 9 displayed point mutations associated with clarithromycin resistance. Of these, 5 samples (55.45%) had the *A2143G* mutation, while the remaining 4 samples (44.45%) had the *A2142G* mutation. Interestingly, two of the samples carrying the *A2142G* mutation were found to be susceptible to clarithromycin according to the E-test method.

Among the PCR-positive samples without any point mutations (28/37), 19 were classified as susceptible to clarithromycin based on the E-test results. Six of the PCR-positive samples were unculturable, and although three samples grew during the initial culture, they could not be propagated further to perform the E-test.

For testing other antibiotics, we successfully cultured 26 *H. pylori*-positive samples. Seven of the samples with a mutation detected by RT-PCR were culturable for the E-test, and 19 of the RT-PCR-positive samples without a point mutation were also culturable. Out of these, 5 samples (19.2%) were resistant to clarithromycin, 6 (23.1%) to metronidazole, 5 (19.2%) to levofloxacin, 1 (3.84%) to amoxicillin, and none showed resistance to tetracycline.

Dilutions of *H. pylori* bacterial load, ranging from 10^6^ CFU/mL to 10 CFU/mL, were performed. *H. pylori* was detected even at 10 CFU/mL, with a Ct value of 42.04.

For *H. pylori* detection, the sensitivity and specificity were 100% and 96.05% (95% Confidence Interval (CI); 93.38–98.73), respectively. For clarithromycin resistance detection, the sensitivity and specificity were 100% and 93.33% (95% CI; 84.4–100), respectively.

These results are summarized in Table 1 and Table 2.

### 3.2. Routine Evaluation

In the second part of our study, 101 of the 502 biopsies (20.11%) were RT-PCR positive, and 90 (17.92%) showed histological evidence of *H. pylori*. RT-PCR detected 11 positive samples (10.89%) missed by histology. These samples, positive by PCR but negative by histology, had a mean Ct of 36.91 while samples positive by RT-PCR and by histology had a mean Ct of 28.87. We detected clarithromycin resistance-associated mutations in 26 (25.74%) of the 101 positive samples; 9 (34.61%) of these patients had no history of previous *H. pylori* eradication therapy.

From the 26 samples positive for a clarithromycin resistance gene, 18 (69.2%) showed the *A2143G* mutation, 7 (26.9%) the *A2142G* mutation, and 2 (7.7%) the *A2142C* mutation. A sample showed both *A2142G* and *A2142C* mutations.

A total of 80 (24.46%) of the adults (n = 327) and 21 (12%) of the children (n = 175) were RT-PCR positive. Of these positive samples, 23 (28.75%) were identified as potentially clarithromycin resistant among the adults, and 3 (14.2%) of positive children’s samples were identified as clarithromycin resistant. Epidemiological results are summarized in Table 3.

For Clarithromycin resistance confirmation and other antibiotic testing, we were able to culture 18 of the 26 *H. pylori* positive samples with a point mutation detected with RT-PCR. All of them are clarithromycin resistant (100%), 9 are Metronidazole resistant (50%), 5 are Levofloxacin resistant (27.77%), 1 is Amoxicilin resistant (5.55%), and no strain showed resistance to Tetracyclin.

## 4. Discussion

The prevalence of *H. pylori*-positive patients in our study population is 20.11%. Despite the lack of recent studies in Belgium on the prevalence of *H. pylori* infections, the values obtained in our study correspond to those reported in various European studies [9,10,11] and in the Belgian NRC report of 2020 (19.4%) [33]. With 25.74% of resistant samples for clarithromycin, we seem to be in accordance with the numbers reported in the European *Helicobacter pylori* Antimicrobial Susceptibility Testing Working Group study (21.4%) [15]. In comparison, Savoldi et al. reported 32% of clarithromycin resistance in the population living in European regions [34].

Adults had a significantly higher prevalence of *H. pylori* than children (24.46% vs. 12%; *p*-value < 0.001). These data are supported by two important meta-analyses by Ren et al. (46.1% (95% CI: 44.5–47.6%) vs. 28% (95% CI: 23.9–32.5%)) [35] and by Chen et al. (43.9% (95% CI, 42.3%–45.5%) vs. 35.1% (95% CI, 30.5–40.1%)) [36]. Furthermore, the higher prevalence of clarithromycin resistance in our adult population is consistent with a meta-analysis conducted by Savoldi et al. on World Health Organization (WHO) regions (28% (95% CI: 25–31%) vs. 24% (95% CI: 19–30%)) [34].

Eight of the 37 positive PCR samples in the first part of our study did not grow on our culture medium. These results are not surprising. Indeed, *H. pylori* culture from gastric biopsy samples typically has a sensitivity of over 90% and a specificity of 100% when performed under optimal conditions [37,38]. The result is only available one to two weeks after sampling, and its accuracy depends on transport and processing conditions [39], such as the skills of the microbiologist, the bacterial load in the biopsy samples, the quality of the samples, the presence of normal microbial flora in the samples, the bacterial load in the gastric biopsy sample, the degree of gastritis (reduction in bacterial load as gastritis progresses), alcohol consumption, bleeding ulcers, use of antibiotics, H2-receptor antagonists, proton pump inhibitors (PPIs), quality and composition of transport media, duration of transport, exposure to air, and transport temperature [37,38,40]. Molecular methods are less affected by storage and transport constraints than culture [15].

In the second part of our evaluation, histology missed 11 (10.89%) positive PCR samples. These results are not surprising compared to several studies [41,42]. Indeed, according to Laine et al., *H. pylori* can be detected by haematoxylin and eosin (H&E) staining with a reported sensitivity and specificity of 69–93% and 87–90%, respectively [41]. Fallone et al. demonstrated an overall accuracy of 92% (95% CI: 86–96%), sensitivity of 93% (95% CI: 87–97%), and specificity of 87% (95% CI: 69–96%) [42].

Specificity can be improved to 90–100% using special stains such as modified Giemsa, Genta, and immunohistochemical (IHC) stains [38,41,43,44]. IHC even provides sensitivity and specificity close to 100% [38]. However, this staining technique is not performed on every sample because it is more expensive and time-consuming [38,41]. That is why the use of IHC is recommended in specific cases such as chronic gastritis, atrophic gastritis, and extensive intestinal metaplasia, in patients treated with PPIs, in follow-up biopsies after eradication treatment, or when *H. pylori* is not identified by histochemical staining [39]. Although histology is recognized as the gold standard method, the results of histology-based methods are influenced by several factors, such as the site of sampling, the size and number of biopsy samples, staining methods, PPIs and antibiotic treatment, pathologist experience, ulcer bleeding [38], and bacterial density [42]. Because PPIs have anti-*H. pylori* activity by reducing the HP load, they can lead to false-negative results. They should be stopped at least 2 weeks before testing. Antibacterial activity of antibiotics and bismuth compounds lasts longer. These drugs must be stopped for at least 4 weeks to allow an increase in a detectable bacterial load [28,31,38,43]. Despite the lack of sensitivity in certain cases, histology has great advantages and must be included in the diagnostic and follow-up algorithm. The advantages of histology include its ability to document *H. pylori* infection, and it also provides more information about the degree of inflammation and associated pathology, such as intestinal metaplasia (IM), atrophic gastritis (AG), gastric cancer, or MALT lymphoma [28,38,39,41].

Compared to other invasive diagnostic methods, RT-PCR offers a rapid and highly reliable diagnosis. Indeed, numerous studies have demonstrated its superiority in terms of sensitivity, specificity, and turnaround time (TaT) [29,45,46].

RT-PCR can help detect *H. pylori* infection in patients with various conditions, such as peptic ulcer bleeding, gastric cancer, or gastric MALT lymphoma, where the diagnosis of *H. pylori* is important but difficult to obtain by other non-molecular methods. PCR methods for *H. pylori* detection have a sensitivity and specificity close to 100% [47,48,49]. This great sensitivity was confirmed in the first part of our study with a positive RT-PCR result even at 10 CFU/mL. Molecular diagnostic methods are less affected by transport conditions and external factors than culture or histology [37,38,40,42]. RT-PCR can detect infection in a significant percentage of histologically negative biopsy specimens. Indeed, according to studies by Bénéjat et al. and Ducorneau et al., RT-PCR shows a 7.5% increase in positivity over bacterial culture methods [29,50]. These studies are consistent with our results.

So far, RT-PCR is considered the most sensitive technique for routine detection of *H. pylori* [37]. However, some false negatives may still occur, which could be reduced by incorporating more sensitive methods. Digital PCR (dPCR), a quantitative technology, offers greater sensitivity compared to conventional or real-time PCR methods while maintaining the same level of specificity [51]. dPCR has proven particularly useful for detecting infectious agents in various sample types and is especially effective in detecting and genotyping resistance genes in *H. pylori* infections [52,53].

Droplet digital PCR (ddPCR) is a technique used to perform dPCR, which relies on generating water-oil emulsion droplets. In ddPCR, the PCR reaction of a sample is divided into thousands of droplets (around 20,000) and subjected to endpoint PCR. Amplification of the target molecules occurs within each droplet, and the positive droplets are detected using a fluorescence reader [54]. The division of a small volume of PCR solution into many droplets enables the absolute quantification of target sequences, which significantly enhances sensitivity compared to other PCR methods [37]. ddPCR is particularly effective in detecting occult *H. pylori* infections, especially in patients with low bacterial density, such as those with peptic ulcer bleeding, gastric MALT lymphoma, or atrophic gastritis, where standard tests may produce suboptimal results [37].

In recent years, clustered regularly interspaced short palindromic repeats (CRISPR)-based nucleic acid detection technology has emerged as a cutting-edge tool for next-generation molecular diagnostics [55]. CRISPR refers to a family of DNA sequences found in prokaryotes, and the associated Cas immune system has gained significant use in molecular biology research. The Cas system targets and cleaves specific nucleic acid sequences, which makes it a powerful tool for genome editing [56]. This method offers remarkable sensitivity. For instance, Yan et al. achieved a limit of detection (LOD) of 1 copy/μL, Qiu et al. reported an LOD of 5 copies/μL, and Wang et al. demonstrated an LOD of 2.2 copies/μL [57].

Clarithromycin resistance due to single mutations such as *A2142C*, *A2142G*, and *A2143G* in the 23S rRNA gene is a major factor contributing to treatment failure of standard clarithromycin-based triple therapy [58]. These mutations have been associated with high levels of clarithromycin resistance worldwide [50,58]. With 25.74% of resistant samples, we have similar results to the European study by Megraud et al., who reported 21.4% of samples resistant to clarithromycin in 2018 [15].

Our findings regarding mutations align with previous studies [59,60,61]. For instance, Chen et al. identified that clarithromycin resistance is predominantly linked to the A2143G mutation (82%), followed by *A2142G* (14%), and *A2142C* (4%) [59]. Similarly, Tamayo et al. observed that resistant *H. pylori* strains harbored the *A2147G* mutation (79.8%), *A2146G* (17.2%), and *A2146C* (2%) [60].

For other antibiotics, during the validation phase, our resistance rates were comparable to those reported in a large European study conducted by the European *Helicobacter pylori* Antimicrobial Susceptibility Testing Working Group. This study found that, after metronidazole, clarithromycin had the highest resistance rates [15], which we discussed in our introduction. The resistance rates presented during routine evaluation are somewhat biased, as additional antibiotics were only tested on samples where a point mutation was detected via RT-PCR.

With increasing clarithromycin resistance rates in *H. pylori*, the use of molecular methods that provide rapid and accurate detection of *H. pylori* infection and clarithromycin resistance simultaneously in real time is being highlighted [38]. Moreover, according to Maastricht VI/Florence guidelines, applying a systematic detection of clarithromycin resistance would allow the use of the optimized triple therapy for 60–90% of patients and therefore limit the consequences of quadruple therapies [28].

In terms of turnaround time (TaT), it is easy to understand that using RT-PCR instead of culture as a first-line diagnostic method is a huge improvement. With RT-PCR, it takes 4 h from the start of manipulation to obtain the results. For culture, the first opening is done after 72 h of incubation and then every 48 h until day 10. In addition, the use of RT-PCR as a first-line diagnosis, with culture used only for E-test testing for strains positive for a mutation associated with clarithromycin resistance, avoids the need to handle and inoculate a large number of negative samples, thus saving laboratory routine time.

## 5. Conclusions

In conclusion, our study confirmed the epidemiological changes we experienced those recent years with an increase in terms of *H. pylori* prevalence and in terms of clarithromycin resistance. The addition of the Allplex *H. pylori* and ClariR Assay in our laboratory routine allowed us to gain much work time, to shorten the TaT, and to be more sensitive than culture and histology combined. Even though histology needs to be maintained for several conditions and culture has to be maintained for other antibiotic susceptibility testing.

## Figures and Tables

**Figure 1 microorganisms-12-02192-f001:**
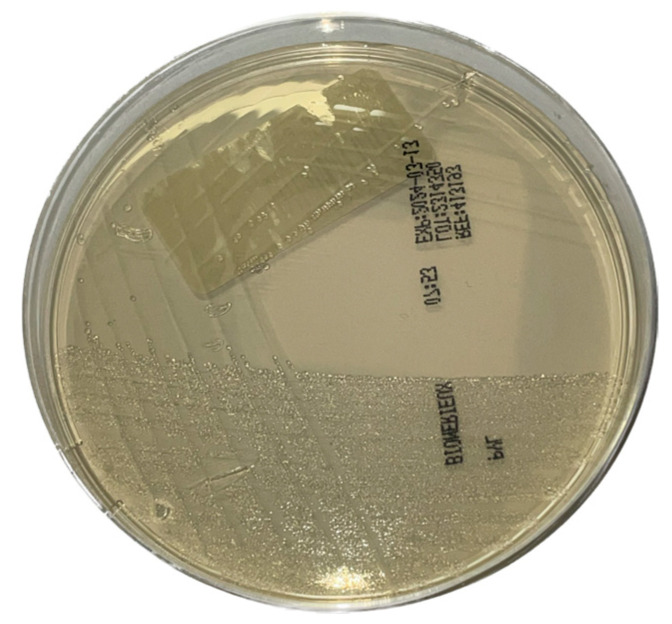
*Helicobacter pylori* culture on bioMérieux^®^ Pylori agar.

**Figure 2 microorganisms-12-02192-f002:**
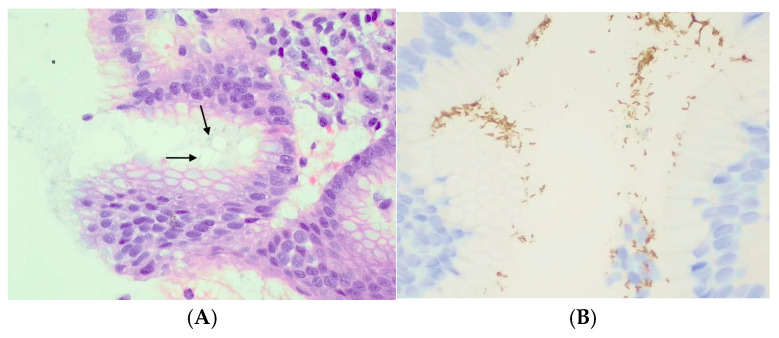
*Helicobacter pylori* seen on gastric biopsy with hematoxylin-eosin staining. Black arrows are pointing *H. pylori* (**A**) and with immunohistochemical staining (**B**).

**Table 1 microorganisms-12-02192-t001:** Results of our method evaluation for HP detection.

Samples (n = 232)	RT-PCR +	RT-PCR −
Culture +	29 (12.5%)	0 (0%)
Culture −	8 (3.4%)	195 (84.05%)

**Table 2 microorganisms-12-02192-t002:** Results of our method evaluation for clarithromycin resistance detection.

Samples (n = 232)	RT-PCR CLR +	RT-PCR CLR −
E-test clarithromycin resistant	7 (18.9%)	0 (0%)
E-test clarithromycin sensitive	2 (5.4%)	28 (75.7%)

CLR: clarithromycin; RT-PCR CLR +: samples with RT-PCR positive for a point mutation associated with clarithromycin resistance; RT-PCR −: samples positive for *H. pylori* RT-PCR without a point mutation associated with clarithromycin resistance.

**Table 3 microorganisms-12-02192-t003:** Epidemiological results of our routine evaluation.

Patients (n = 502)		RT-PCR +	RT-PCR −	CLR Resistance Associated Gene(s)
Age	0–16 (n = 175; 34.86%)	21 (12%)	154 (88%)	3 (14.2%)
	17–85 (n = 327; 65.13%)	80 (24.46%)	247 (75.54%)	23 (28.75%)
Gender	Men (n = 218; 43.43%)	48 (22.01%)	170 (77.9%)	11 (24.91%)
	Women (n = 284; 56.57%)	53 (18.66%)	231 (81.34%)	15 (28.38%)
Total		101 (20.11%)	401 (79.89%)	26 (25.74%)

CLR: Clarithromycin.

## Data Availability

The data presented in this study are not available due to the General Data Protection Regulation (GDPR) in our institution. This protocol was reviewed by the Research Ethics Committee (EC Research) of the CHC Mont-Légia (OM087). All diagnostic procedures were performed routinely. All patients were examined in a hospital setting, according to good clinical practice, with informed consent for the endoscopic procedure, followed by appropriate treatment if necessary. In this routine process, consent for endoscopic procedures is always given in writing and recorded in the patient’s medical record. Informed consent for the use of human gastric DNA was not sought from patients. Therefore, to ensure the anonymity of the subjects, all indirectly identifiable patient data were removed from the present study.

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
