# Peer review of "Detection by Real-Time PCR of Helicobacter pylori and Clarithromycin Resistance Compared to Histology on Gastric Biopsies"

_microorganisms, 2024, doi:10.3390/microorganisms12112192_

Round 1
Reviewer 1 Report
Comments and Suggestions for Authors
The manuscript “Detection by real-time PCR of Helicobacter pylori and clarithromycin resistance compared to histology on gastric biopsies” by G. Pittie and coauthors carried out a study to validate the implementation of a kit in routine laboratory use for the diagnosis of Helicobacter pylori and the 3 main mutations associated with clarithromycin resistance. Furthermore, the authors compare, in 2 different phases, the results of RT-PCR against bacterial culture and against histology. The results, as expected, reflect a greater sensitivity and specificity of RT-PCR.
The study highlights the importance of improving the sensitivity, specificity and turnaround time in the diagnosis of HP and its resistance to clarithromycin, which can be published. However, the study has moderate relevance since it is specifically based on the use of a commercial brand kit with expected and little novel results.
Moreover, there are various flaws in the planning and execution of the procedures, as well as omission of details in laboratory and statistical methods used. On the other hand, the manuscript is significantly condensed and needs to be consistently improved in all its sections. Below I list specific observations for these sections:
Abstract
· The abstract presents significant inconsistencies. In the presentation of the methods, the authors go directly to the second part of the study (routine evaluation) while in the results they present the sensitivity and specificity values ​​that were calculated in the first part of the study (method validation). Then again, they show results from the second part. The abstract must clearly reflect that there were 2 parts and each one with its respective most relevant results.
· Lines 20-21: The conclusion highlight the value of a specific commercial brand. This may reflect the existence of potential conflicts of interest that should be addressed in a different disclosure than that presented at the end of the manuscript.
Introduction
· Line 27: The scientific name Helicobacter pylori should be written in italics whenever it is mentioned throughout the manuscript.
· Line 32: The acronym used HP must be detailed at length at its first mention in the manuscript, regardless of the abstract.
· Line 34: I recommend being consistent in mentioning HP or H. pylori and using only one form throughout the manuscript.
· Lines 40-41: This information is wrong and must be corrected. The document cited literally says: "Approximately 89% of non-cardia gastric cancer cases, representing 78% of all gastric cancer cases, are now estimated to be attributable to chronic H. pylori infection."
· Lines 43-46: I recommend listing other antimicrobial resistances reported in H. pylori (metronidazole, amoxicillin, rifampicin, levofloxacin, etc.) and their epidemiological context to contextualize the importance of clarithromycin resistance.
· Lines 58-60: I suggest adding additional information on the mechanisms of clarithromycin resistance. Which gene(s) are involved, the number and type of known mutations, the individual impact of each mutation on the degree of resistance (since sites 2142 and 2143 do not have the same effect).
Methods
· Section 2.1.: Why didn't the authors use the same samples to compare the 3 methods? The authors point out that they compare PCR vs bacterial culture (n=232) and then PCR vs histology (n=502) in different samples (since they are from different time periods). This seems like a flaw in the planning phase. In addition, they must provide more details for both sample groups: age, sex, injury, etc.
· Since the authors mention 2 groups of analysis, indicate from the beginning of sections 2.2 to 2.5 in which and how many samples these procedures were performed.
· Section 2.5.: The numbering of the sections is out of order. I suggest adding some details about these procedures. For example, were the samples pretreated before DNA extraction? Furthermore, the last sentence should come immediately after mentioning the assay, before referring to the analyses. On the other hand. In this section and whenever reference is made to the assay used in this study, refer accurately: real-time PCR or RT-PCR (see inaccuracy in line 77 for example).
· Line 96: correct initial dilution number to exponential.
· Line 100: What about clarithromycin? it is not listed here.
· Section 2.4.: This section needs more details. What does "if required" imply? Was there any criterion in the study to decide between conventional staining with hematoxylin-eosin and immunohistochemical staining? What was the number of samples processed with each type of staining.
Results
· Line 122: If the authors show the results of the first analysis (PCR vs bacterial culture) why do they mention results of the second analysis (PCR vs histology?). This is confusing and again demonstrates failure in planning. Explain these results better or remove them if they were not uniformly performed in the 232 biopsies.
· Line 123: Of the 9 HP positive for mutations, detail which mutations were found in the 7 resistant HP and in the 2 susceptible HP. This is important to show which mutations have greater penetrance.
· Line 124: add "potentially" or "are associated with"... resistance to clarithromycin, since as the study itself shows, some HP with mutations were susceptible to CLR.
· The authors performed E-test to evaluate resistance against Levofloxacin, Metronidazole, Tetracycline and Rifampicin. Indicate the results.
· Lines 130-132: Provide more details on how the sensitivity and specificity analysis were carried out in the methods.
· Table 1: Column headers should be RT-PCR + , or RT-PCR - for CLR.
· Lines 136-137: Why did the authors place this addendum here instead of placing it from the first mention on line 122? The analyses were performed twice?. And again, why are there histological results for the first group of biopsies?
· Table 2: I suggest using the acronym CLR to refer to clarithromycin. Describe the meaning in the legend.
· Section 3.2.: Based on the results shown in 3.1. It is clear that being RT-PCR positive for mutations associated with resistance to CLR does not mean that they are positive in phenotype. The authors should appropriately refer to "potentially" or "associated mutations" to CLR resistance. On the other hand, the authors must evaluate the phenotypic resistance in those 26 isolates and report what those 26 mutations were.
· Line 142: If the kit is not designed to detect specifically H. helmanii why do the authors say they missed a positive sample?
· Line 142: Helicobacter helmanii should be in italics.
· Line 144: What about the mean Ct for PCR + and histology +? Detail also that.
· Line 145: change to “resistance-associated mutations”
· Lines 148-149: Children are not resistant to clarithromycin, but HP can be.
· Table 3: “age” title should not be in bold neither underlined. “CLR resistance-associated mutations” or a legend describing that. The notes in the legend should be also shown in Tables 1 and 2 when appropriate.
Discussion:
· The discussions must be carried out in the same order that the results were derived. First the results on the validation of the method and then those derived from the routine evaluation.
· Line 156: prevalence of what?
· Again, 20.11% represents mutations associated with resistance to CLR and not the phenotypic resistance (which is most important). Update this discussion after performing the CLR E-test on those 26 isolates.
· Lines 161-162: What is the point of this random comparison without discussion? The way it is mentioned raises more questions than comparisons: genetics of American people? climate? year of study? ages of the population studied? Extend the importance of these comparisons.
· Line 164: What statistical test did the authors perform to establish this p-value?
· Lines 165-168: Detail the prevalence values ​​reported in the cited studies.
· Lines 182: Detail the range of values ​​reported in the cited studies.
· I agree with the discussion in paragraphs 4, 5 and 6. However, it should be considered that "so far" RT-PCR methods can be considered the most sensitive in routine detection. However, it is possible that there are still false negatives that can be improved with the implementation of more sensitive methods (for example ddPCR, isothermal amplification + CRISPR) or better detection protocols. Add a discussion about it.
· 7° paragraph: This discussion should be extended based on the differential contribution of the 3 mutations to the resistance phenotype, after having also detailed them in the results. Furthermore, the authors must address the discrepancies or provide explanations about why the kit detected 2 strains as resistant to CLR while the culture classified them as susceptible.
Conclusions:
· Lines 235-236: To justify this part of the conclusion (which must be maintained), it is necessary to show the results of the other antibiotics evaluated in 2.3.
Others
· Acknowledgments section should be updated.
· Since patient samples were studied, an ethical approval statement must be added.
Reviewer 2 Report
Comments and Suggestions for Authors
This manuscript provides a study on detection by real-time PCR of Helicobacter pylori and clarithromycin resistance compared to histology on gastric biopsies. This study involved human gastric biopsies obtained from patients submitted to endoscopy for various gastric conditions. Therefore, authors should provide the ethical approval with Ethical code No. and the informed consent written by the patients/participants. This manuscript lacks a comprehensive and detailed summary of current methods for testing Helicobacter pylori in “Introduction”. Especially, there are no specific measurement results in this study, and the data provided are all summative, so the authenticity of the data needs to be verified. In addition, this manuscript lacks sufficient experimental procedures and valid information. Finally, there are many details that should be noticed and this manuscript requires tighter editing. It is unfortunately that this manuscript is unsuitable for publication on Microorganisms.
The specific comments are listed in detail as follows:
1. This study involved human gastric biopsies obtained from patients submitted to endoscopy for various gastric conditions. Therefore, authors should provide the ethical approval with Ethical code No. in this manuscript. The informed consent written by the patients/participants should be provided in this study.
2. In the section of “Introduction”, this manuscript lacks a comprehensive and detailed summary of current methods for testing Helicobacter pylori.
3. In this study, HP and clarithromycin mutations were analyzed using real-time PCR, but the specific operational information of PCR was missing.
4. For Figure 1, the plate is placed in an anaerobic environment for culture, but how does the plate ensure an anaerobic environment?
5. CI appears many times in the text, but what is CI? The author should provide enough information.
6. Table 1 is not clearly expressed and it is not clear which methods compare the results.
7. There are no specific measurement results in this study, and the data provided are all summative, so the authenticity of the data needs to be verified. It is suggested that the author put all the test results of the actual samples into the supporting material.
8. The Latin names of microorganisms in this manuscript are not written correctly. For example, the Latin name of the microorganism “Helicobacter pylori” should be italicized.
9. There are some details that should be noticed and this manuscript requires tighter editing. For example,
1) Line 96, “106 CFU/mL to 10 CFU/mL”
2) Line 103, the legend in Figure 1 is at the top of the diagram
3) The reference format does not meet the journal requirements
Reviewer 3 Report
Comments and Suggestions for Authors
The manuscript microorganisms-3240187 presents an original investigation on the usefulness of PCR assays to detect Helicobacter pylori and clarithromycin resistance in gastric biopsies compared to classical histopathology and immunohistochemical procedures. The main aim is the evaluation of a possible superiority of PCR in improving laboratory performance, therefore sensitivity and specificity are reported for the two tested phases, i.e. method validation and routine evaluation.
Due to the global burden of H. pylori infection and antibiotic resistance and given the importance to administer an appropriate therapy for H. pylori infection, this work is of interest and deserves publication. The paper is generally well written and clear, and the results are clearly presented. I have only minor suggestions to improve the methodological and the conclusion sections, as follows.
Whole text. Decimal values. Please check the text for uniformity (e.g., % are usually reported with commas, p-values with dots)
Methodological section. Please provide more information on patients’ characteristics, including (but not limited to) how many children/adults and the indication for biopsies. Also provide information on ethical approval and informed consent.
Methodological section. Please include more details on DNA extraction and Real-time PCR procedures with Allplex and ClaryR assay kits. If the kits were used according to manufacturer's instruction, please specify and/or include suitable references
Methodological section. How many pathologists were involved in the evaluation of histology on a same sample?
Methodological section. Please add a paragraph for the description of statistical analyses.
Results. Line 122. Specify the meaning of Ct
Conclusions. Can the authors provide more information on the work time and turnaround time analysis, and elaborate more on that?
Round 2
Reviewer 1 Report
Comments and Suggestions for Authors
I have reviewed the first version of the manuscript "Detection by real-time PCR of Helicobacter pylori and clarithromycin resistance compared to histology on gastric biopsies" and each response to the observations in this second version. Thank you for taking the care to review the observations in detail. The authors have satisfactorily answered or justified all of them and the new version is considerably improved. One last observation is to remember that every time you mention H. pylori it should be in italics (I noticed this in some of the new lines added).
Reviewer 2 Report
Comments and Suggestions for Authors
I have evaluated the revised version of the manuscript (microorganisms-3240187). Authors have carefully evaluated the reviewers’ and editor’s comments and suggestions, responded to the suggestions point-by-point, and revised the manuscript accordingly. I think that the revised version of the manuscript (microorganisms-3240187) is suitable for publication on Microorganisms.